# The Judicious Use of Stereotactic Ablative Radiotherapy in the Primary Management of Localized Renal Cell Carcinoma

**DOI:** 10.3390/cancers15143672

**Published:** 2023-07-19

**Authors:** Andrew B. Barbour, Simon Kirste, Anca-Liga Grosu, Shankar Siva, Alexander V. Louie, Hiroshi Onishi, Anand Swaminath, Bin S. Teh, Sarah P. Psutka, Emily S. Weg, Jonathan J. Chen, Jing Zeng, John L. Gore, Evan Hall, Jay J. Liao, Rohann J. M. Correa, Simon S. Lo

**Affiliations:** 1Department of Radiation Oncology, University of Washington, Fred Hutchinson Cancer Center, Seattle, WA 98195, USA; 2Department of Radiation Oncology, Medical Center, Faculty of Medicine, University of Freiburg, German Cancer Consortium (DKTK) Partner Site Freiburg, 79085 Freiburg, Germany; 3Division of Radiation Oncology and Cancer Imaging, Peter MacCallum Cancer Center, University of Melbourne, Parkville, VIC 3052, Australia; 4Department of Radiation Oncology, Sunnybrook Health Sciences Centre, University of Toronto, Toronto, ON M5S 1A1, Canada; 5Department of Radiology, School of Medicine, University of Yamanashi, Yamanashi 409-3898, Japan; 6Division of Radiation Oncology, Juravinski Cancer Centre, McMaster University, Hamilton, ON L8V 5C2, Canada; 7Department of Radiation Oncology, Cancer Center and Research Institute, Houston Methodist Hospital, Houston, TX 77030, USA; 8Department of Urology, University of Washington, Fred Hutchinson Cancer Center, Seattle, WA 98195, USA; 9Department of Medical Oncology, University of Washington, Fred Hutchinson Cancer Center, Seattle, WA 98195, USA; 10Department of Radiation Oncology, London Health Sciences Centre, London, ON N6A 5W9, Canada

**Keywords:** kidney cancer, radiation oncology, RCC, renal cancer, renal carcinoma, SABR, SBRT, stereotactic body radiation therapy

## Abstract

**Simple Summary:**

Patients with localized renal cell carcinoma often have medical comorbidities limiting their surgical candidacy, thus necessitating less invasive treatment options. Stereotactic ablative radiotherapy has emerged as a safe and effective management option with a growing body of evidence supporting its use. This article discusses recent advances in the use of stereotactic ablation radiotherapy for localized renal cell carcinoma, while guiding providers on practical points for patient selection and clinical application.

**Abstract:**

Localized renal cell carcinoma is primarily managed surgically, but this disease commonly presents in highly comorbid patients who are poor operative candidates. Less invasive techniques, such as cryoablation and radiofrequency ablation, are effective, but require percutaneous or laparoscopic access, while generally being limited to cT1a tumors without proximity to the renal pelvis or ureter. Active surveillance is another management option for small renal masses, but many patients desire treatment or are poor candidates for active surveillance. For poor surgical candidates, a growing body of evidence supports stereotactic ablative radiotherapy (SABR) as a safe and effective non-invasive treatment modality. For example, a recent multi-institution individual patient data meta-analysis of 190 patients managed with SABR estimated a 5.5% five-year cumulative incidence of local failure with one patient experiencing grade 4 toxicity, and no other grade ≥3 toxic events. Here, we discuss the recent developments in SABR for the management of localized renal cell carcinoma, highlighting key concepts of appropriate patient selection, treatment design, treatment delivery, and response assessment.

## 1. Introduction

The incidence of renal cell carcinoma (RCC) is rising, potentially due to increased incidental detection by medical imaging [1,2]. Globally, renal cancers comprise approximately 2.2% of new cancer cases and 1.8% of cancer deaths [3]. The peak incidence of RCC occurs between the 7th and 8th decade of life, and over a third of new cases occur at age 75 years or later [1]. In addition to advanced age, common risk factors for RCC include smoking, obesity, and hypertension [1]. Thus, many patients diagnosed with RCC are at an increased risk of complications from anesthesia and surgical resection, highlighting the critical need for effective, less-invasive definitive treatment options.

Historically, RCC has been considered a radioresistant tumor due to poor response to conventional fractionation in preclinical studies [4]. However, in vitro cell cultures demonstrated that RCC cells could be ablated by high fractional radiation doses [5], and higher radiation doses were found to be effective for palliation of advanced stage RCC [6,7]. These findings were followed by advancements in radiation treatment techniques allowing for ablative doses of radiation to be delivered in a highly conformal and accurate manner. Highly conformal treatment delivery is particularly important in the treatment of primary RCC given the radiosensitivity of adjacent organs at risk (OARs), most notably the luminal gastrointestinal organs.

Stereotactic ablative radiotherapy (SABR), synonymous with stereotactic body radiotherapy (SBRT), refers to highly conformal delivery of ablative doses of radiation (generally >6 Gy per fraction) to extracranial sites of disease. In SABR, the planning target volume (PTV), which includes tumor and a margin accounting for patient setup uncertainty, is covered with an ablative dose, while a higher dose is intentionally delivered heterogenously within the PTV. For example, a PTV prescribed 10 Gy to the 80% isodose line will be covered with a minimal dose of 10 Gy, but may have a dose of 12.5 Gy (10 Gy/80%) or greater within the target. Dose heterogeneity within the PTV allows for sharp dose fall-off outside of the PTV in order to spare OARs, while creating a central high-dose region that may help overcome radioresistance from hypoxic conditions [8]. Delivery of such sharp dose gradients requires daily image guidance, advanced motion management techniques, and strict quality-assurance procedures in order to ensure treatment accuracy on the order of millimeters [9].

Given the widespread adoption of SABR and the perceived sensitivity of RCC to high fractional doses, multiple centers have begun employing SABR for the primary treatment of localized RCC with encouraging initial outcomes [10,11,12,13,14,15,16]. Herein, we will discuss the recent developments in SABR for the management of localized renal cell carcinoma, highlighting key concepts of appropriate patient selection as well as radiation treatment design and delivery.

## 2. SABR Rationale and Patient Selection in Localized RCC

### 2.1. Overview of Staging and Management Options

Initial staging requires multiphase abdominal imaging and chest radiography [17]. Abdominal computed tomography (CT) with contrast is typically performed for T-staging. Magnetic resonance imaging (MRI) is useful for patients that cannot tolerate CT contrast agents or when evaluating local invasion [17]. Contrast-enhanced ultrasound is useful for clarification of indeterminate findings, most notably complex cystic masses [18]. Tumors limited to the kidney are classified by size criteria as T1 (T1a ≤ 4 cm, T1b > 4 and ≤7 cm) or T2 (T2a > 7 and ≤10 cm, T2b > 10 cm). Tumors extending into perinephric tissues or major veins, but bounded by Gerota’s fascia, are classified as T3 regardless of size, while those extending beyond Gerota’s fascia or to the ipsilateral adrenal gland are T4. For patients without lymph node involvement or distant metastatic disease (TxN0M0), American Joint Committee on Cancer (AJCC) 8th edition prognostic group staging is determined by T-stage (T1 = Stage 1, T2 = Stage 2, T3 = Stage 3, T4 = Stage 4) [19].

For localized RCC (TxN0M0), primary management options rely upon T-stage, tumor location, and patient factors such as renal function, comorbidities, and perceived operative risk. Per societal guidelines, surgical resection (partial vs. radical nephrectomy) is the preferred management option [20,21]. For patients declining surgery, deemed to be poor surgical candidates, or who are medically inoperable, active surveillance and ablative techniques remain as options, particularly for patients with small T1a tumors [22,23]. If proceeding with active surveillance or ablative therapy, a biopsy is recommended to confirm the histologic diagnosis prior to treatment and guide post-treatment surveillance [21].

Under the paradigm of active surveillance, small renal masses may be initially monitored with the option of treatment for progression following a process of shared decision-making. Active surveillance of small, low-grade RCCs is justified based upon low risk of metastatic spread and significant competing risks of death in older patients with comorbidities [24,25,26]. Although no randomized clinical data exists for active surveillance, a systematic review of 28 studies showed a 1–6% risk of metastatic progression among patients on active surveillance over 24–93 months of follow-up [27]. Patient age, tumor size, and growth rate may serve as risk factors for metastatic progression [28]. In patients aged ≥ 75 years, surgical management of cT1 tumors may not increase survival compared with surveillance but accelerates renal dysfunction, which associates with cardiovascular mortality [26]. A national cancer database population-based cohort study showed an increased risk of death for patients with T1N0M0 kidney cancer managed with observation as opposed to surgery, ablative therapy, or SABR, but interpretation of this work is highly confounded by selection bias [29]. In addition to tumor size, tumor growth, and patient comorbidities, histologic subtype may help inform discussions surrounding active surveillance [30]. A recently published risk calculator may help personalize treatment selection for patients with T1 renal cortical masses [31].

Thermal ablative therapies include modalities such as cryotherapy, radiofrequency ablation, and microwave ablation. Although less invasive than surgical resection, thermal ablative therapy requires laparoscopic or percutaneous access, which is associated with risk of periprocedural morbidity in highly comorbid patient populations. Within properly selected patients, these treatments may have a 90% or greater clinical efficacy [32,33,34,35,36]. However, treatment with these techniques is generally limited to smaller (<3–4 cm) T1 tumors spatially distant from the hilum and proximal ureter [21] and is recommended to be cautiously offered due to remaining uncertainties regarding oncologic efficacy [37]. There are no randomized studies to help guide selection of treatment modality when patients are candidates for both thermal ablation and SABR, although the RADSTER trial (NCT03811665) has completed accrual and may help clarify such situations. As a general guideline, thermal ablation may be initially preferred in patients with bilateral tumors or a genetic predisposition to develop multiple tumors that may require additional future tumor-directed therapies [21,38].

### 2.2. Societal Guidelines and Rationale for SABR in Localized RCC

Per version 4.2023 of the National Comprehensive Cancer Network (NCCN) practice guidelines for kidney cancer, SABR may be considered for medically inoperable patients with Stage I (category 2B evidence) or Stage II–III (category 3) cancer. European Association of Urology (EAU) guidelines broadly consider ablative therapies as an alternative to surgery for treatment of older patients with small renal masses and comorbid patients deemed unfit for surgery [21]. EAU guidelines state that ablative therapies should not be used for tumors larger than 3–4 cm, while focusing mainly on thermal ablative techniques. EAU guidelines cite a single source reviewing the role of nephron-sparing treatment techniques in T1 tumors [39], citing no other studies of SABR in localized RCC. European Society of Medical Oncology (ESMO) guidelines do not include SABR as a treatment option for localized RCC, instead stating that SABR may be used in the advanced/metastatic setting to treat local disease in poor surgical candidates who are not eligible for other local therapies such as thermal ablation [38,40].

As prospective trials evaluating the use of SABR for localized RCC have had small cohort sizes [11,12,13,41], the most robust data evaluating its efficacy comes from pooled analyses. A systematic review and meta-analysis evaluated local control (LC), toxicity, and renal function of 372 patients (383 primary tumors) from 26 studies, including 11 prospective trials, with a median follow-up of 28 months [10]. Mean tumor size was 4.6 cm (range 2.3–9.5) and mean pre-SABR estimated glomerular filtration rate (eGFR) was 59.0 mL/min (range 28.7–89.8). Weighted random-effect models estimated LC at 97.2% (95% confidence interval [CI] 93.9–99.5), grade 3–4 toxicities at 1.5% (95% CI 0–4.3%), and eGFR change as −7.7 mL/min (−12.5 to −2.8). Notably, this meta-analysis included patients with metastatic disease who underwent SABR to the primary tumor, but at least 80% of patients had localized RCC.

Recently, a multi-institutional individual patient data meta-analysis of 190 patients with localized RCC treated with SABR published results with a median follow-up of 5 years [16]. This study included adult patients regardless of performance status, while excluding patients with metastatic disease, prior abdominal radiotherapy, prior upper tract urothelial carcinoma, or with contraindications to renal SABR. Of the cohort, 96 of 128 patients with available data regarding the treating physician’s assessment of their operative candidacy were deemed inoperable, including 17 due to existing or anticipated renal dysfunction. Fifty-six patients had a solitary kidney and no patients received adjuvant or concurrent systemic therapy. Pre-SABR median tumor size was 4 cm (interquartile range [IQR] 2.8–4.9) and pre-SABR eGFR was 60.0 (IQR 42.0–76.0). The primary endpoint of local failure (LF) was evaluated radiographically. The 5-year cumulative incidence of LF and distant failure with death as a competing risk were 5.5% (2.8–9.5) and 10.8% (6.6–16.2), respectively. During the follow-up period, 66 patients died, including ten cancer-related deaths (five from other malignancies). Grade 1–2 treatment-related toxicities were recorded in 37% of patients, but only one patient experienced grade ≥3 events. In the five years following treatment, eGFR decreased by a median 14.2 mL/min (IQR 5.4–22.5), including seven patients who progressed to requiring dialysis.

### 2.3. Patient Evaluation Prior to SABR

Following diagnosis, staging, and surgical evaluation, multiple factors must be considered to optimize patient selection for primary treatment of localized RCC with SABR. Patient history must be evaluated for factors that would potentially increase the toxicity risk of abdominal SABR, such as prior abdominal radiotherapy, inflammatory bowel disease [42], connective tissue disorders [43], or concurrent administration of certain systemic therapies [44]. Due to the unique management implications of disorders such as Von Hippel-Landau (VHL) disease, patients should be considered for genetic testing if presenting with multiple renal masses, age ≤ 46 years at diagnosis, or with a family history of RCC [20,45]. A prospective study of SABR in seven patients with VHL who had multiple bilateral cysts, refused surgery, or could not undergo nephron-sparing resection demonstrated 2-yr LC of 100% with no grade ≥2 toxicities and minimal effect of eGFR after a 43-month median follow-up period [46]. As patients with VHL often have disease that is multicentric, bilateral, and prone to recurrence, multidisciplinary evaluation should precede treatment to ensure an optimal management strategy that may include a serial combination of local therapies.

For patients with bilateral kidneys, a nuclear medicine split-function perfusion scan (i.e., renogram) should be obtained to assess relative function of the involved and uninvolved kidney. All patients require evaluation of baseline renal function. Creatinine-based GFR estimating equations are recommended for routine clinical use [47]. The Chronic Kidney Disease Epidemiology Collaboration (CKD-EPI) equation for GFR estimation may most accurately classify patients across a broad range of populations [48]. Studies have reported variable estimates of eGFR decline following SABR, likely due to heterogeneity in tumor size, tumor location, and treatment techniques. Broadly, eGFR may be expected to decrease in the order of 5–15 mL/min in the years following SABR [10,11,14,16,46]. Larger tumor size may correlate to greater functional loss, potentially due to a larger area of high-dose overlapping functional parenchyma and less sparing of normal renal cortex [14,49,50]. However, some of this decline may be due to a patient’s natural trajectory of eGFR unrelated to SABR [11], and some patients may have an increase in eGFR following treatment [51]. Although a clear eGFR cutoff for SABR candidacy is not defined, exclusion of patients with eGFR < 30 mL/min can be considered to reduce the risk of iatrogenic dialysis [52], although some institutions selectively treat below this cutoff after assessment of bilateral renal function. For patients with a solitary kidney, SABR can often be safely employed after evaluation of baseline eGFR [49].

Tumor characteristics also affect patient selection for SABR. As there is little data on radiation for non-clear cell histologies, it is not clear if tumor histology is an important consideration for SABR treatment decisions. Although there is no clearly defined upper size limit for lesions that can be managed with SABR, safe and efficacious treatment has been clearly demonstrated for T1b tumors [51,53]. For patients with tumors > 7 cm, a retrospective study of 11 patients (9 of whom had nodal or distant metastatic disease) demonstrated that SABR to the tumor alone vs. whole kidney was well tolerated [54]. Until additional data becomes available regarding size criteria, T2—particularly T2b—tumors may serve as a reasonable size-based exclusion criterion for patient selection [52]. Tumor extension into neighboring veins, such as the inferior vena cava (IVC), increases the risk of morbidity and mortality from resection and upgrades tumor stage to T3 [55]. Although surgery remains the only curative option in the presence of IVC tumor thrombus, a retrospective multi-institutional study of SABR for IVC tumor thrombus found thrombus regression in 7 of 12 patients with post-SABR imaging, showing SABR is at least a palliative option for T3 disease [56]. Ongoing work is evaluating the use of neoadjuvant-SABR of IVC tumor thrombus [57,58]. Finally, tumor location relative to radiosensitive OARs must be assessed, as direct contact of tumor with bowel complicates safe delivery of SABR.

## 3. Treatment of Primary RCC with SABR

### 3.1. Selection of Treatment Prescription and Normal Tissue Constraints

There is no consensus regarding the optimal dose and fractionation for primary management of RCC with SABR. The α/β ratio of RCC is unclear, as in vitro studies have suggested a ratio between 2.6 and 6.92 Gy [5]. If assuming an α/β ratio of 3, a biologically effective dose (BED) of BED_3_ > 225 Gy may provide optimal rates of control [59]. Higher doses have been delivered without dose-limiting toxicities. However, in a study of 11 patients treated over three fractions to either 48, 54, or 60 Gy (BED_3_ = 460 Gy), progressive disease was only documented at the highest dose level [12], potentially indicating a lack of benefit for continued dose-escalation. Given prior data on safety and efficacy, as well as the relative iso-effectiveness of these doses over the range of plausible α/β ratios, an ongoing clinical trial selected 42 Gy in three fractions (BED_3_ = 238 Gy) for tumors > 4 cm and 26 Gy in 1 fraction (BED_3_ = 251 Gy) for smaller tumors [52]. A similar dose of 48 Gy in four fractions (BED_3_ = 240 Gy) has also been used safely [60]. For current clinical practice, 42 Gy in three fractions can be recommended as a well-tolerated regimen with high rates of local control. For tumors ≤ 4 cm, 26 Gy in one fraction can be considered since single-fraction treatment may provide superior oncologic outcomes and patient convenience as compared to multi-fraction regimens [16]. For large T2 tumors, or those in close proximity to bowel, 35–40 Gy in five fractions (BED_3_ = 117–147 Gy) may be used to limit toxicity risk, although doses up to 50 Gy in five fractions (BED_3_ = 217 Gy) should be attempted when feasible due to the lower control rates seen with lower doses [11,54]. Ideally, doses should be prescribed to the 70–80% isodose line, but this practice is not standardized and published work has not consistently reported treatment isodose [52,61].

Normal tissue dose constraints can influence selection of prescription dose and fractionation, particularly for tumors in close proximity to small bowel. As exemplified by a cohort of 190 patients managed with SABR [16], maximum point dose constraints should be strictly enforced to avoid high-grade toxicity of the small bowel. Among that cohort, one grade 4 duodenal ulcer was documented, which occurred in a patient receiving a small bowel point dose of 54 Gy in four fractions [16]. To further mitigate risk of gastrointestinal toxicity, an every-other-day schedule should be considered for multi-fraction regimens [62]. Caution should be used when treating with a single fraction regimen in close proximity to the ureters, particularly in the setting of a solitary kidney, due to the paucity of ureteral toxicity data for single-fraction treatment [63,64]. Dose to the ipsilateral kidney should be as low as reasonably achievable (ALARA), minimizing the volume receiving a high fractional dose (approximately > 50% isodose) [50]. As shown in Table 1, various OAR constraints have been used in clinical practice, and a standardized set of dose-constraints for the treatment of localized RCC does not exist.

### 3.2. Informed Consent

Patient consent for treatment should include fatigue and nausea, the most common side effects, as well as the possibility of dermal changes, chest wall or flank pain, mildly decreased renal function, and gastrointestinal effects such as diarrhea or gastritis. Radiation may cause tissue scarring that can complicate future surgical intervention, although most patients treated with SABR are already poor operative candidates. Very rare risks depend upon a patient’s specific situation and include gastrointestinal damage that may require surgery, spinal cord injury, and renal failure. One case of acute intratumoral hemorrhage, presenting with flank pain, fever, and vomiting, has been reported [66]. Radiation-associated secondary malignancy is a theoretical risk, but no cases have been reported in the literature following treatment for localized RCC. Overall, SABR for RCC is generally well tolerated with a minority of patients experiencing treatment-related toxic events [16].

### 3.3. Treatment Setup, Design, and Delivery

Various treatment delivery systems have been employed for renal SABR, such as tomotherapy, gantry-mounted, robotic, and MRI LINAC [61,67]. Patient simulation techniques vary by treatment platform and institutional preference, but patients will generally be in a supine position with arms up [61]. Patients should not be positioned directly on the treatment couch. Instead, a patient positioning device, such as a negative pressure vacuum bag, should be used for reproducibility of setup and patient immobilization. Generally, CT simulation should be performed with intravenous (IV) contrast for patients without absolute contraindications. If IV contrast cannot be given at time of simulation, a diagnostic contrast-enhanced scan should be co-registered for tumor delineation. CT simulation slice acquisition thickness should be 1–3 mm.

A variety of appropriate imaging techniques can be used to define the gross tumor volume (GTV). Typically, the GTV will be contoured on the simulation CT and further verified via use of a co-registered diagnostic MRI. Positron emission tomography (PET) scans can also be co-registered for GTV delineation, but the utility of PET is limited by physiologic radiotracer excretion leading to PET-avidity of normal renal parenchyma, thereby decreasing contrast between normal tissue and PET-avid renal lesions [68]. However, PET may have increased utility when delineating tumor thrombus for the treatment of T3 disease [68], and novel techniques such as PSMA-1007 PET may better delineate tumor from renal background due to more prominent hepatic excretion [69]. The use of single-photon emission CT (SPECT) has also been described for GTV delineation, and SPECT may be particularly useful for patients that cannot tolerate CT or MRI contrast [70]. Expansions for creation of a clinical target volume (CTV) are not routinely recommended [14,16,52,57,67,71].

Motion management is essential given the effects of respiratory motion on renal position, and multiple appropriate motion management methods exist [72]. Four-dimensional CT (4DCT) is commonly used to accumulate GTV positions throughout all respiratory phases to create an internal target volume (ITV). The ITV should not be defined as the full extent of motion between deep inspiration and deep expiration. Instead, an additional motion management technique, such as abdominal compression or respiratory gating, should be coupled with 4DCT. A reasonable goal of advanced motion management is to limit ITV expansion to no greater than 0.5 cm axially and 1.0 cm craniocaudally [57]. The need for an ITV may be obviated via the use of real-time, intra-fractional imaging, such as with MRI LINAC or robotic tracking of internal fiducials [16,67,71]. On-board image guidance is required prior to each treatment fraction to ensure proper patient and tumor positioning. Cone beam CT (CBCT), with or without fiducial markers, is commonly used for image guidance. If not using fiducial markers, CBCT should be used to align to the ipsilateral kidney. CBCT should be repeated during the treatment fraction if there are concerns about patient movement during treatment, which may be monitored via optical surface imaging. Margins for PTV are dependent upon treatment platform, form of image guidance, and institutional practice, generally ranging from 0–5 mm [16]. Generally, the PTV will be trimmed off luminal gastrointestinal structures and the PTV prescription goal will be ≥95–99% coverage with 95–100% of prescribed dose [16,52,61].

Options for treatment beam arrangement depend foremost upon the delivery system. For example, ring-gantry systems may be limited to isocentric coplanar beams, while robotic systems use non-isocentric, non-coplanar beams. When using a gantry-mounted LINAC, patients may be treated with static intensity modulated radiotherapy (IMRT) or dynamic volumetric modulated arc therapy (VMAT). Non-opposing, non-coplanar beams are preferable for high dose sparing of OARs and may consist of ≥6 static IMRT beams or 2–5 VMAT arcs. For example, beam arrangement may include 6 static co-planar IMRT beams and 1–2 non-coplanar IMRT beams. However, if needing to prioritize functional renal parenchyma sparing due to presence of solitary kidney or due to results of split-function perfusion scan, coplanar beams are preferable, such as the use of two VMAT arcs (Figure 1).

### 3.4. Treatment Response Assessment

Imaging follow-up for local response is typically conducted with CT or MRI, and radiographic assessment of localized RCC response to SABR remains a clinical challenge [61]. RCC tumors grow slowly and exhibit slow radiographic response to radiotherapy. In a retrospective study of 41 tumors managed with SABR, mean pretreatment linear tumor growth rate was 0.68 cm per year compared to −0.37 cm per year after treatment with no statistically significant change in tumor enhancement [15]. In addition to persistent enhancement after treatment, some tumors may show an initial period of radiographic growth prior to regression [15,56,73]. Therefore, response criteria for thermal ablation, where persistent enhancement or failure of size regression are considered as signs of local recurrence, cannot be applied to SABR [74]. Instead, Response Evaluation Criteria in Solid Tumors (RECIST) criteria can be useful, but caution should be practiced when interval growth is detected on short-interval post-treatment scans. Determination of response assessment may be postponed to 1-year follow-up [52]. As SABR is not expected to induce a complete radiographic response of primary RCC, local success should be judged by rates of stable disease and partial response [17]. Ongoing work on novel PET tracer ligands, such as PSMA-1007, may lead to additional options for evaluation of therapeutic response assessment [69,75].

Post-SABR biopsy has been employed but is not recommended as a routine measure of response assessment because of its inherent ambiguity and potential risk for leading to unnecessary salvage procedures [76]. Residual histologic disease is routinely identified on biopsies conducted 6–12 months after treatment, but finding residual tumor cells does not appear to relate to local failure in longer term follow-up, likely due to post-treatment tumor senescence [11,12,13]. If post-treatment biopsy is conducted, advanced pathologic techniques may be considered to evaluate for cellular senescence among surviving tumor cells [11]. Due to the unique biologic treatment mechanisms of SABR, residual tumors cells are likely to have lost their proliferative potential and would thus not be considered viable. Additional techniques, such as cell-free DNA, have yet to be validated for response assessment [77,78].

## 4. Conclusions and Future Directions

For patients with localized RCC who desire treatment but are not eligible for surgical management or thermal ablative therapies, non-invasive SABR provides high rates of local control coupled with a low toxicity profile [10,16]. In patients eligible for active surveillance, SABR, and/or thermal ablative therapy, we lack clear evidence guiding optimal management strategy, although evidence supporting SABR may be more rigorous than for thermal ablation [16,37]. Until better evidence is available, (e.g., reporting of results from the RADSTER trial NCT03811665), shared decision-making after multi-disciplinary consultation should be employed in these situations [31]. Although T1 and T2a tumors can be safely managed with SABR, further work is needed to refine upper size criteria. T3 tumors may be treated in the palliative setting with SABR, and ongoing work is investigating SABR of T3 tumors in the neoadjuvant setting [56,57,58]. Societal guidelines should be updated to include SABR as a management option for select T1, T2a, and T3 tumors, depending on patient-specific factors and treatment intent. An optimal eGFR threshold for SABR candidacy does not exist, but exclusion of patients with eGFR < 30 mL/min should be considered to reduce the risk of iatrogenic dialysis.

Future work is needed to refine optimal selection of dose-fractionation, the effect of isodose prescription, and to understand if single fraction treatment yields superior outcomes compared with multifraction regimens. Various published OAR dose constraints exist, but these have not been standardized or refined in the setting of localized RCC. However, even with a diverse set of dose-constraints used in clinical practice, treatment-related toxicity has been minimal. An optimal strategy for treatment response assessment remains to be defined and is a limitation confounding post-SABR surveillance. Ongoing work with advanced imaging techniques, such as multiparametric MRI with diffusion weighted imaging and dynamic contrast enhancement, may advance the treatment response assessment paradigm [52].

## Figures and Tables

**Figure 1 cancers-15-03672-f001:**
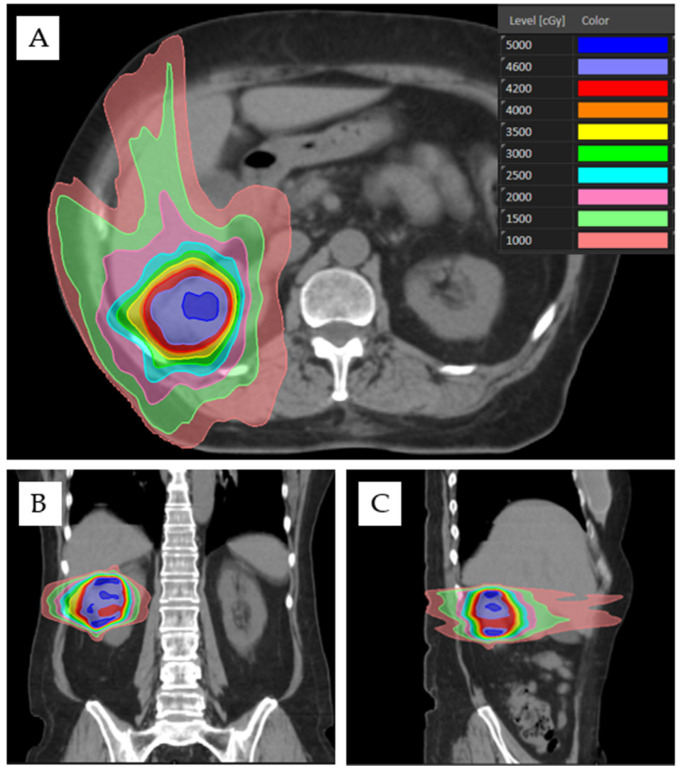
Treatment of a right-sided WHO grade 2 clear cell RCC (cT1aN0M0) with two 184-degree VMAT arcs of 6 MV energy. A negative pressure vacuum bag, abdominal compression, and 4DCT were used for motion management. The patient was prescribed SABR as 42 Gy in three fractions to the 75.6% isodose line, obtaining 100% PTV coverage. OAR metrics included: D0.035cc to bowel (19.0 Gy), stomach (17.1 Gy), spinal canal (6.4 Gy), and contralateral kidney (2.0 Gy); V10Gy to contralateral kidney (0%); D700cc to liver (0.7 Gy); and 37% of the ipsilateral kidney covered by the 50% isodose line. (**A**–**C**) Axial, coronal, and sagittal CT views with color-coded radiation dose distribution in centigray (cGy).

**Table 1 cancers-15-03672-t001:** Representative normal tissue dose constraints for one, three, and five fraction SABR.

Organ	One Fraction	Three Fractions	Five Fractions
	D0.035cc < 22 Gy ^a^ or 26 Gy ^b^	D0.035cc < 30 Gy ^a,b^	D0.035cc < 35 Gy ^a^ or < 29 Gy ^d^
Small bowel/duodenum	D5cc < 17.4 Gy ^a^ or < 22.5 Gy ^b^	D5cc < 22.5 Gy ^a^ or D30cc < 12.5 Gy ^b^	D5cc < 26.5 Gy ^a^
	Maximum dose to full bowel wall circumference ≤ 12.5 Gy ^b^		
Large bowel	D0.035cc < 31 Gy ^a^ or D1.5cc ALARA, aim for < 26 Gy ^b^	D0.035cc < 45 Gy^a^ or D1.5cc ALARA, aim for <42 Gy ^b^	D0.035cc <52.5 Gy ^a^ or <29 Gy ^d^
Stomach	D0.035cc < 22 Gy ^a^ or D1.5cc < 15.4 Gy ^b^	D0.035cc < 30 Gy ^a,b^	D0.035cc <35 Gy ^a^ or <29 Gy ^d^
Liver	D700cc < 11.6 Gy ^a^	D700cc < 15 Gy ^b^ or <17.7 Gy ^a^	D700cc <19.6 Gy ^a^ or D50% < 25 Gy ^d^
Ipsilateral kidney–ITV	ALARA, minimize volume of >50% IDL ^b^	ALARA, minimize volume of >50% IDL ^b^	D60% < 15 Gy ^d^
Contralateral kidney	V10Gy ≤ 33% ^b^	V10Gy ≤ 33% ^b^	D100% < 11 Gy ^d^
Ureter	D0.035 cm^3^ < 35 Gy ^a^	D0.035 cm^3^ < 40 Gy ^a^	D0.035 cm^3^ < 45 Gy ^a^
Spinal canal	D0.035 cm^3^ < 12 Gy ^c^	D0.035 cm^3^ < 18 Gy ^c^	D0.035 cm^3^ < 27.5 Gy ^c^

^a^ Timmerman 2022 [65]. ^b^ Siva et al. 2018 [52]. ^c^ Siva et al. 2016 [61]. ^d^ Lapierre et al. 2023 [60]. The authors do not assume responsibility for use of these dose limits, and not all constraints have been thoroughly tested.

## Data Availability

Data sharing not applicable.

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
