# Peer review of "The Judicious Use of Stereotactic Ablative Radiotherapy in the Primary Management of Localized Renal Cell Carcinoma"

_cancers, 2023, doi:10.3390/cancers15143672_

Round 1

Reviewer 1 Report

Authors should be congratulated for their work. The topic is interesting and intriguing. In a society where the average age is increasing and with it the incidence of renal masses in the elderly, it is essential to pay attention to the possible treatments to be used in these scenarios. However, the current review does not add any novelty compared to other very recent ones (PMID: 35843777; 36383304).

Author Response

We thank the reviewer for their comments and agree that this topic is of emerging importance. The reviewer's sole criticism is one of novelty, and two recently published reviews are provided to highlight their critique (Ali et al 2022 & Rich et al 2022).

Ali et al. 2022 broadly reviewed the use of SABR in RCC, including in the metastatic setting. That review dedicated approximately 1 page to localized RCC. Rich et al. 2022 provides a more in-depth review of SABR for localized RCC, but lacks many aspects our review addresses.

Since the time of those reviews, new evidence has emerged (e.g., Siva et al. 2022, Glicksman et al 2023, Hannan et al 2023) that is incorporated into our review. Compared to both reviews, our review provides additional details surround the management of localized RCC. For example our review provides extensive detail on patient selection, alternative treatment options, and details of SABR design and delivery that are not part of those papers, such as CT simulation, OAR doses, beam design, and follow-up. Thus we believe our work represents a substantially different and significant contribution.

Reviewer 2 Report

Robot-assisted partial nephrectomy (RAPN) is standard of care for small RCC and achieved excellent clinical outcomes. However, RAPN is sometime difficult to perform, because of poor general condition, elderly, and tumor location.

SABR as well as thermal ablation are promising alterative treatment options.

This review focuses on the important clinical issue and provides valuable information for physicians.

Author Response

We thank the reviewer for their comments and we agree that our review focuses on an important clinical issue and provides valuable information for physicians.

Reviewer 3 Report

The review by Barbour et al. “The Judicious Use of Stereotactic Ablative Radiotherapy in the Primary Management of Localized Renal Cell Carcinoma” provides an updated analysis of applying precision-guided radiation therapy for primary renal carcinoma. It appears that ablative radiation therapy can be effective for early stage renal cancer in patients who are not candidates for other invasive procedures. RCC is generally considered as radioresistant, and more recent experimental results have not changed that view, even though RCC cells in vitro can certainly be killed by radiation in the form of physical damage, therefore the line 57 regarding the rationale of using radiation for RCC requires more deliberation. For the discussion of future direction regarding the treatment response assessment, as these RCC is slow growing, the imaging analysis may be complemented by other non-invasive means such as measurement of ctDNA, urine cfDNA as more sensitive methods of gauging tumor growth. 

Author Response

We thank the reviewer for their comments on our updated analysis on the use of precision-guided radiation therapy for primary RCC. Our responses to the reviewers two comments are as follows:

  • Regarding the reviewer's comment on the radioresistance of RCC, we have updated the line in question to read: "Historically, RCC has been considered a radioresistant tumor...".
  • Regarding the reviewer's comment on response assessment, we have added the text: "Additional techniques, such as cell-free DNA, have yet to be validated for response assessment[77,78]."

Round 2

Reviewer 1 Report

Authors should be congratulated for their work. The revisions made in response to my inquiries were meticulous and left no stone unturned. It is with great pleasure that I acknowledge their exceptional efforts in thoroughly addressing the concerns I had previously raised. Based on their comprehensive responses, I am delighted to affirm that the manuscript is now suitable for publication in its current form.